# Development of a Cranial Suture Traction Therapy Program for Facial Asymmetry Correction Using the New Delphi Technique

**DOI:** 10.3390/medicina58070869

**Published:** 2022-06-29

**Authors:** Seong-Yeon Park, Hea-Ju Hwang, Kyu-Nam Park

**Affiliations:** 1Majoring in Public Health, Department of Medicine, General Graduate School, Cha University, Seongnam-si 13503, Korea; hlspasoo6366@naver.com; 2Majoring in Medical Beauty Industry, Graduate School of Public Health Industry, CHA University, Seongnam-si 13503, Korea

**Keywords:** cranial traction therapy, correcting facial asymmetry, New Delphi technique

## Abstract

*Background and Objectives**:* We aimed to develop a cranial suture traction therapy program, a non-surgical therapeutic method for facial asymmetry correction. *Materials and Methods:* Six experts, including rehabilitation medicine specialists, oriental medical doctors, dentistry specialists, five experts, including Master’s or doctoral degree holders in skin care and cosmetology with more than 10 years of experience in the field, 4 experts including educators in the field of skin care, a total of 15 people participated in the validation of the development of the cranial suture traction therapy program in stages 1 to 3. Open questions were used in the primary survey. In the second survey, the results of the first survey were summarized and the degree of agreement regarding the questions in each category was presented. In the third survey, the degree of agreement for each item in the questionnaire was analyzed statistically. *Results:* Most of the questions attained a certain level of consensus by the experts (average of ≥ 4.0). The difference between the mean values was the highest for the third survey at 0.33 and was the lowest between the second and third surveys at 0.47. The results regarding the perceived degree of importance for each point of the evaluation in both the second and third stages of the cranial suture traction therapy program were verified using the content validity ratio. The ratio for the 13 evaluation points was within the range of 0.40−1.00; thus, the Delphi program for cranial suture traction therapy verified that the content was valid. *Conclusions:* As most questions attained a certain level of consensus by the experts, it can be concluded that these questions are suitable, relevant, and important. The commercialization of the cranial suture traction treatment program will contribute to the correction and prevention of facial dislocations or asymmetry, and the developed treatment will be referred to as cranial suture traction therapy (CSTT).

## 1. Introduction

In modern society, many individuals want to improve their image by changing their appearance, regardless of sex or age. Changes in their external appearance in the direction that others prefer by various methods such as cosmetic surgery, makeup, and skin care can stimulate one’s self-confidence [1,2]. Furthermore, as competition intensifies, it is becoming harder to determine providers’ competency; therefore, the elements of discrimination, which are more obvious and easier to perceive, are becoming one of the more important evaluation criteria. The beauty trend is dominated by small and slim faces with clear and healthy skin. Hence, cosmetic plastic surgery has attracted increasing attention [3,4].

The face is a sensitive structure, and even minor changes in life can result in distortion of the bones supporting the skeleton and the facial muscles. In particular, if facial asymmetry is not corrected on time, it can lead to secondary problems not only in appearance but also in other parts of the body, leading to increased concerns about disease. Facial asymmetry is considered a negative factor for an individual’s image.

The major cause of facial asymmetry due to lifestyle is the excessive formation of masticatory muscles [5], as well as facial swelling, double chin, excessive fat, inconspicuous facial contours, drooping cheeks, and sagging jaw lines [6]. Additionally, various congenital or acquired factors, such as unilateral mandibular hyperplasia, hemifacial microsomia, mandibular condyle fractures, and condylar ankyloses, can cause facial asymmetry [7,8]. Ectopic mandibular symphysis is known to be important in identifying facial asymmetry [9,10].

Looking at the causes of facial asymmetry in the mechanical structure of the human body, humans are organic cubes consisting of multi-directional stereotactic structures that act in gravity. The two axes of the spine core, left and right, constantly change the organic function phases [11]. In everyday life, the powerful force of gravity affects the entire fascia, causing deformations of the skull, spine, and sacrum [12]. While humans walk and run, the muscles from the tip of their toes to the head systematically relax and contract [13]. According to the general pattern of diagonal walking on two feet, when the right shoulder faces the front, the upper arm bone is accompanied by external rotation and the lower arm bone is flexed. The forearm is supinated, and the wrist joint forms radial deviation compensation movements. When the pelvis is facing forward in the leg, the femur is shaped in the form of outer rotation and flexion, and a compensation movement to the toes is made [14,15,16]. At this point, the neck rotates in the direction of the curved shoulder to form a compensatory movement to the face. As such, each part of the human body is a dynamically connected structure, which results in considerable compensation and has a significant impact on facial displacement [17].

Habits repeated during one’s lifetime result in the formation of an unbalanced or distorted face shape with asymmetry between both sides or the front and rear parts of the face [18]. Furthermore, turtle neck, spinal scoliosis, and pelvic slope heavily affect the body balance. Body imbalance is difficult to restore by itself; thus, symptoms will continue to worsen [19,20].

Symptoms of facial asymmetry can lead to temporomandibular joint disorders, such as temporomandibular joint pain, articulation, and trismus disorders, and may present with ophthalmodynia, cephalalgia, painful neck, and leg length imbalance [21,22]. Nevertheless, people with facial asymmetry do not easily recognize the changes in their faces and their seriousness [23].

Facial asymmetry refers to an incomplete match in size, location, and shape of bilateral structures in relation to the centerline [24]. As most people have this asymmetry, it has been referred to as one of the morphological characteristics of the body [25,26]. Lum et al. reported that features of facial asymmetry were found even in normal people [27]. Nonetheless, the number of patients visiting a clinic due to facial asymmetry is increasing. Thus, a more systematic and accurate approach to facial asymmetry is necessary.

The assessment of soft and hard tissues of the frontal face is critical for the diagnosis, prediction, and evaluation of treatment in facial asymmetry [5]. Although facial asymmetry is diagnosed through quantitative evaluation of hard tissues, it is recognized not only by the asymmetry of hard tissues but also by that of soft tissues [28,29]. Yogosawa suggested that skeletal abnormalities can be covered by soft tissues, such as muscles and skin [30]. Duran et al. [31] and Shamlan and Aldrees [24,32] reported that people with symmetrical facial features still have skeletal asymmetry and proved that there is a difference in the degree of asymmetry between hard and soft tissues. Ferrario et al. [33] and Haraguchi et al. [34] identified differences between the hard and soft tissues in terms of the role of asymmetry in recognizing class III asymmetry of the craniofacial skeleton.

The cranium grows up to over 90% of the adult size by the age of 5–7 years. The facial bones grow in an anterior–posterior direction before and in a posterior direction after the age of 3 years. At the age of 8 years, there is a shift from horizontal to vertical growth. Thereafter, the cranium grows in various types and directions depending on the time and area, with different proportions of the facial and cranial parts of the head [35,36,37]. Normal growth results in the ratio between the face and the cranium being even, achieving a harmonious appearance even if the left and right may not be identical. However, in some cases, the harmony between the face and cranium can be broken with a malfunction resulting from a defect in the normal process or an abnormal development during formation and growth [38].

Cranial suture traction therapy is a therapeutic method in which the suture surfaces of cranial bones are pushed or pulled, keeping in mind that the craniofacial skeleton has its own movements. It increases each movement of the craniofacial skeleton and changes the frame of the otherwise limited cranial and facial bones [39]. It is a therapy that applies adequate time and intensity to effect changes in the skeleton by the direction of the unique movements of the cranial and facial bones to treat facial asymmetry, according to the principle of craniosacral therapy [40]. In particular, as the movements of the cranial bones affect the facial bones, suturing the cranium plays an important anatomical role in correcting facial asymmetry [41]. Tariq et al. reported that as the angle of the cranial base increases, the condylion’s position shifts toward the posterior upper part [42]. By looking at the resulting movement of the mandible rotating posteriorly and upward, it can be predicted that the change in the cranial bones forming the base of the cranium will be related to facial asymmetry.

Matsushita et al. reported that facial asymmetry is indicated by differences in the vertical length of the left and right structures or the horizontal width of the midline [43]. However, classification by skeletal shape type revealed that the differences in types have significance. The skeletal shape could appear differently depending on the motion limitation of the cranial or facial bones; therefore, in their study, they tried to examine whether the skeletal shape would be measured differently after improving the limitations in the components of each bone. In addition, Duran et al. suggested that a difference in the degree of asymmetry between soft and hard tissues was seen in patients with facial asymmetry, indicating that an analysis of asymmetry using items of the soft tissues’ measurement would be necessary along with that of the hard tissues for evaluation of facial asymmetry [31].

In the field of skin care and cosmetology, a wide range of studies on facial reduction have examined various methods, such as meridian massage, tibia massage, miso facial acupuncture, skeletal muscle therapy, and electrical nerve stimulation [44,45,46,47]. However, studies evaluating facial asymmetry through soft or hard tissue in the area of skin care are scarce. Therefore, an effective facial correction method should be designed to avoid time-consuming and costly cosmetic surgery procedures and reduce the burden of side effects, providing treatment through a relatively simple procedure.

The purpose of this study was to develop a cranial suture traction therapy program, which is a non-surgical therapeutic method that can be employed to help correct facial asymmetry of hard tissues through the treatment of soft tissues. We propose a scientific and systematic program to be developed using point-by-point stimulation manual therapy on the cranial bone suture surface, a method that is currently being used in orthopedics, oriental medical center, and therapy shops.

## 2. Materials and Methods

### 2.1. Study Subjects and Design

The purpose of this study was to develop a cranial suture traction therapy program for correcting facial asymmetry by conducting a study that considers the opinions of various experts. Accordingly, a suitable method of choice was the Delphi technique, so that content validity for the development of the cranial suture traction therapy program can be verified.

Regarding the Delphi technique, the selection of subjects was critical to the success of the study. The panel of experts chosen was as follows: rehabilitation medicine specialists (2), oriental medical doctors (2), dentistry specialists (2), educators in the field of skin care in accordance with the Higher Education Act (4), and master’s or doctoral degree holders in skin care and cosmetology with more than 10 years of experience in the field (5). These experts were selected based on whether they had made previous presentations on the topic of cranial suture traction therapy or for their career experiences in their respective positions as researchers in their institutes.

Table 1 presents the number of subjects who participated in the survey over the three stages. Prior telephone or face-to-face contact was conducted to present the purpose of the study. Experts who agreed to participate were selected. The survey started with the first 15 subjects, a number that was selected based on the theories of Adler and Ziglio [48] and Giannarou and Zervas [49]. A final group of 15 people participated in the third stage. To meet the targeted number of opinions to be collected from experts during the second and third stages, the study comprised three stages and it was verified that there were no experts left out from contributing their opinions in all three stages. The first survey began in June 2020, and the third survey was completed on 30 July 2020.

#### 2.1.1. Delphi Survey Tools

The Delphi technique allows for the systematic introduction and contrast of perceived judgments on a specific subject [50]. The first step in this process is the selection of experts in the relevant field. To increase the validity and credibility of the Delphi results, the following characteristics of the experts must be considered: representativeness, relevance, expertise, integrity of participation, and adequacy in number [48,49]. Based on this theory, 15 experts were selected to participate in the Delphi survey.

The number of experts to be selected for the panel can vary to meet the needs and size of the study, but a sample group of at least 10 people must be formed in order for errors in the mean values to be minimized and for the study to be credible. In general, samples between 10 and 15 are accepted as valid for the Delphi survey [50]. The Delphi technique systematically analyzes and synthesizes the responses given by the group of experts to predict, diagnose, and solve certain problems until the goal of group consensus is achieved.

#### 2.1.2. Selection of Development Tools

Table 2 presents the 14 items for cranial suture traction therapy selected by William Sutherland, a student of Still (the founder of osteopathic medicine on the human brain), based on the concept of manually dealing with cranial structures [51]. The principle of biomechanics of all joints in the human body is a passive and active system of the joints that distributes forces to achieve balance between movement and restriction; it is a system of balance that controls the action of movements. The selection based on the active system included joints other than those that brought about movements.

### 2.2. Theoretical Background of the Selected Development Tools

Similar to other joints in the human body, the joints of the cranium have movements that result from external as well as internal forces that bring about internal balance. Some of the typical features of the joint structure include maintenance of its structure by fixation of the restorative ligaments, free nerve terminal, veins and arteries, and proprioceptors in sutures, such as the slippery side, superior periosteum, falx cerebri, and tentorium cerebri [52].

Regarding the biomechanics of cranial motion, first, due to bone plasticity, minute movements of cranial bones are possible along the anterior and posterior axes and the horizontal and vertical axes. The occipital, sphenoid, frontal, ethmoid, and vomer bones mostly have movements of flexion and extension. The movements of the temporal, parietal, palatine, and zygomatic bones are made up of external and internal rotation [40].

The general movement of the cranial system is a cyclic movement in the active and passive stages, which is a combination of external rotations that result from the flexion and expansion of the ossa centrale; the passive stage is a combination of internal rotations that result from the extension of the ossa centrale and the relaxation of the surrounding bones. Each bone shows a specific movement along the three axes [52]. The movement of the occipital bone is caused by bending and extending around the horizontal axis located at the intersection of the two sides that pass the anterior boundary of the greater occipital bone and the upper boundary of the occipital bone.

The movement of the sphenoid bone occurs downward and forward with respect to the central axis located at the center of the sphenoid bone by its greater wing and along the horizontal axis passing through the sphenoid bone rostrum. The movement of the anterior forehead bone moves the posterior and upper boundary of the anterior forehead bone posteriorly and downward around the central axis, and the eyebrow bow moves forward and upward.

The movement of the temporal bone has three axes of rotation: the petrobasilar pivot that meets the occipital and sphenoid bones, the sphenopetrous pivot, and the petrojugular pivot that meets the occipital and sphenoid bones. The petrobasilar pivot slides so that the cranium can be expanded, and the petrojugular pivot causes an outer rotation of the horizontal axis and a counterclockwise rotation of the vertical axis. The sphenopetrous pivot rotates around the clinoid process insertion point of the petrosphenoid bone ligament.

The movement of the parietal bone occurs where the coronal suture meets the frontal bone and the squamous suture meets the temporal bone, with the lambdoid suture meeting the occipital bone with outer rotational movement that bends simultaneously. The outer part of the coronal suture is moved forward and everted. The inner part of the coronal suture and the depressed bregma are moved posteriorly. The sagittal suture surface is depressed with the border of the parietal bone and moves posteriorly.

The lambdoid suture becomes depressed, and the lambdoid suture surface of the posterior head boundary moves backward. The squamous suture surface becomes depressed, and the outer part of the parietal bone moves forward in an ectropion position [53].

Second, the movement of cranial bones transfers movement to all bones that make up the maxillary bone and joints by delivering the movement of the sphenoid, temporal, and frontal bones to the maxillary bone [39,52].

The movement of the maxilla is the inner and outer rotational motion of the maxillary bone, where the upper part passes through the front of the maxillary bone, and the lower part moves through the anterior lateral angle at the slanted axis. When the sphenoid bone is bent, the maxillary bone moves posteriorly as the suture between the maxillary bones is lowered, and each maxillary bone moves backward. The posterior boundary of the frontal bone is raised, and the anterior surface moves anterolaterally.

The movement of the zygomatic bone forms the joints with the maxillary, frontal, and temporal bones, rotates anterior–outward during bending, pulls the orbital boundary of the zygomatic bone towards the outside, allowing the orbit to widen. The frontal bone protrusion moves to the anterior and lateral surfaces, and the temporal process of the zygomatic bone moves downward–outward [53].

In addition, based on the findings of Ferrario et al. [33] and Haraguchi et al. [34], the 14 items of cranial suture traction therapy were set as shown in Table 3.

### 2.3. Research Procedure

In the first stage, evaluation papers about cranial suture traction therapy were given, and an example of the said therapy was presented to hint towards the correct answers for this test. The questionnaire consisted of the participant’s consent form, personal information, the method of data collection, necessity, purpose, and terms to provide a clear understanding. The application duration, frequency, and intensity were composed of 10 s per point, 3 times of application frequency, and 2 kg/cm^2^ of application intensity through validation of the contents of 5 related experts based on a study by Park [54]. The open-ended were questions related to the 14 items on cranial suture traction therapy. The content analysis of the survey was conducted by five experts to ensure reliability in conducting this analysis: a rehabilitation medicine specialist, a doctor of a Chinese medical college, and three professors from the Department of Skin Care and Cosmetology. They organized the results related to their respective fields. As a result of the analysis of the first open-ended questions, a total of 14 items were selected for the cranial suture traction therapy program for facial asymmetry correction. Table 2 summarizes the components of the inspection questionnaire.

The second survey reflected the opinions collected from the first survey. We created a structured close-ended questionnaire through consultation with experts, namely two rehabilitation medicine specialists, two oriental medical doctors, two dental specialists, five master’s or doctoral degree holders in skin care and cosmetologies with more than 10 years of experience in the field, and four educators in the field of skin care. Subsequently, five advisors (a rehabilitation medicine specialist, a doctor of a Chinese medical college, and three professors from the Department of Skin Care and Cosmetology) were required to evaluate the validity of the survey. We presented the collected data and process of opinions from the first survey, and the validity was directly indicated by the expert group on a five-point Likert scale. In addition, we revised, deleted, added, and presented the opinions on the structured contents by the first survey, if necessary.

In the third Delphi survey, we discussed the response results of the second Delphi survey that consisted of similar questions through consultation with experts, namely two rehabilitation medicine specialists, two oriental medical doctors, two dental specialists, five master’s or doctoral degree holders in skin care and cosmetologies with more than 10 years of experience in the field, and four educators in the field of skin care. The validity of each question was determined by five advisors (a rehabilitation medicine specialist, a doctor of a Chinese medical college, and three professors from the Department of Skin Care and Cosmetology) and indicated using a five-point Likert scale, similar to the second Delphi survey.

The second and third surveys were conducted via email. The second set of questions included those that were made through content analysis of the first open-ended questions. Next, they were finalized using a 5-point Likert scale, as follows: 5 points for “very suitable”, 4 points for “somewhat appropriate”, 3 points for “normal”, 2 points for “slightly unsuitable”, and 1 point for “very unsuitable”. The mean and standard deviation of the items were acquired. According to the questions of each category, items with a level of consensus (average of ≥4.00) were selected.

In the results of the first survey, out of the 14 items, the level of consensus (average ≥ 4.0) of experts was low with the fourth item regarding the frontal and nasal bone traction question not achieving consensus. This question was deleted, and the second and third surveys were conducted with 13 question items.

The third survey consisted of the same questions as those in the second stage, except that they were marked with the results from the second survey: the opinions of each individual, along with the average of each question. Consequently, as shown in Table 4, 13 questions were developed, and the results from the second and third surveys revealed that all questions had a high level of consensus (average ≥ 4.0) among the experts. A study on the cranial suture traction therapy program for facial asymmetry correction was developed, consisting of a total of 13 items.

In this study, the items for traction therapy, which was conducted according to the designated numbers of each cranial suture, were classified according to Table 4, the number of treatment points for the traction therapy ranged from 1 to 3, and the processes of change of the parts were converted into data, keeping each of the treatment points separate.

### 2.4. Processing Data and Statistical Analysis

The answers collected at each stage were checked for any omissions and entered into the computer one by one in the form of data that could be analyzed. IBM SPSS Statistics v.23.0 (IBM Corp., Armonk, NY, USA) was used to apply the New Delphi technique to the results of the second and third surveys and calculate the frequency distribution and average difference between the results, as well as their X^2^. The level of significance was set at *p* < 0.05.

Based on the consolidation of the responses from experts, the means, standard deviations, medians, and interquartile ranges were calculated. In addition, to determine the validity of each component, the content validity ratio (CVR) and degree of consensus and convergence were calculated. The CVR was calculated using the formula developed by Lawsh [49]:(1)$$\mathrm{CVR}=\frac{N_{e-\frac{N}{2}}}{\frac{N}{2}},$$
where *N* = number of responses and $N_{2}$ = the sum of the number of ‘very important’ and ‘important’ responses according to the Likert scale.

The degree of consensus and convergence must be calculated to determine whether the opinions collected from a panel of experts have reached a consensus. To calculate this statistic, median and quartile values were needed. *Q*1 and *Q*3 represent values corresponding to the first and third quartile coefficients, 25% and 75%, respectively. The following formulas were used to calculate the degree of consensus and convergence [55,56]:$$Consensus=1-\frac{Q3-Q1}{Mdn}Convergence=\frac{Q3-Q1}{2}$$

## 3. Results

The results of the development of the cranial suture traction therapy program for facial asymmetry correction are shown in Table 5 and Table 6.

In the second survey, the highest mean value was >4.60 and related to the questions regarding traction of the frontal and maxillary bones, traction of the frontal and sphenoid bones, traction of the sphenoid and zygomatic bones, traction of the zygomatic and temporal bones, and traction of the zygomatic and maxillary bones. The lowest mean value was 4.40, which related to the question regarding the traction of the frontal and parietal bones.

In the third survey, the highest mean value was 4.93, which related to the questions regarding traction of the frontal and sphenoid bones, traction of the frontal and zygomatic bones, traction of the frontal and parietal bones, traction of the parietal and sphenoid bones, traction of the parietal and temporal bones, traction of the occipital and temporal bones, traction of the sphenoid and temporal bones, traction of the sphenoid and zygomatic bones, traction of the zygomatic and temporal bones, and traction of the zygomatic and maxillary bones. The lowest mean value was 4.87, which related to the questions regarding traction of the frontal and parietal bones, traction of the frontal and maxillary bones, and traction of the parietal and occipital bones.

The difference between the mean values was the highest for the third survey, a value of 0.33, and the lowest between the second and third surveys, a value of 0.47. However, because most of the questions attained a certain level of consensus by the experts (average of ≥4.0), it can be said that most are suitable and important.

The results regarding the degree of importance for each of the points of evaluation made by the groups of experts in both the second and third stages of the cranial suture traction therapy program were verified using the CVR. The ratio for the 13 points of evaluation was within the range of 0.40−1.00; thus, the content of the cranial suture traction therapy program was verified as valid.

Table 5 shows the results of analyzing the validity of the evaluation factors (Consensus, Convergence, etc.) for the results of the cranial suture traction treatment program in the second survey.

Table 6 shows the results of analyzing the validity of the evaluation factors (Consensus, Convergence, etc.) for the results of the cranial suture traction treatment program in the third survey.

Table 7 shows the adopted results for questions that did not show adequate levels of significance (*p* < 0.05) by verifying the homogeneity (credibility).

## 4. Discussion

The purpose of this study was to develop a cranial suture traction therapy program to correct facial asymmetry. In order to do this, the cranial suture traction therapy was suggested by a group of experts, and by testing its validity and credibility, a cranial suture traction therapy program that could be used to correct facial asymmetry was developed.

As the suture lines of the cranium are small amounts of connective tissue interposed between the bones, thereby connecting them, each part of the cranium can be divided into individual parts [51]. Through this division, this study established and categorized each item of cranial suture traction therapy.

The treatment program of this study ultimately follows that mentioned in the book Cranial Osteopathy (Principles and Practice) by Liem et al. [39] in that it performs traction at the parts that the book deemed effective for correction therapy. These include the frontal bone spread technique involving the frontal bone part and the part where traction of the frontal and parietal bones was performed in this study (item #1); the cant hook technique involving the frontal bone part and the part where traction of the frontal and maxillary bones was performed in this study (item #2); the alternative technique involving the frontal bone part and the part where traction of the frontal and sphenoid bones was performed in this study (item #3); the frontozygomatic cannot hook technique involving the part where traction of the frontal and zygomatic bones was performed in this study (item #5); the parietal spread technique involving the parietal bone part and the part where traction of the parietal bones was performed in this study (item #6); the parietal technique involving the sphenoid bone part and the part where traction of the parietal and sphenoid bones was performed in this study (item #7); the lambdoid suture technique involving the temporal bone part and the part where traction of the parietal and temporal bones was performed in this study (item #8); the lambda technique involving the occipital bone part and the part where traction of the parietal and occipital bones was performed in this study (item #9); the occipitomastoid suture direct technique involving the occipital bone part and the part where traction of the occipital and temporal bones was performed in this study (item #10); the sphenosquamous pivot technique involving the sphenoid bone part and the part where traction of the sphenoid and temporal bones was performed in this study (item #11); the sphenozygomatic technique involving the zygomatic bone part and the part where traction of the sphenoid and zygomatic bones was performed in this study (item #12); the temporozygomatic bone technique involving the zygomatic bone part and the part where traction of the zygomatic and temporal bones was performed in this study (item #13); and the zygomatic bone alternative technique involving the zygomatic bone part and the part where traction of the zygomatic and maxillary bones was performed in this study (item #14). Thus, the treatment program developed in this study will prove effective.

The validity and credibility of the development of this program were verified by an expert, and it was ascertained that all the items were effective in bringing about facial symmetry and are useful items to be applied in alternative therapy. In addition, the treatment points were chosen according to the stimulation methods used for traction therapy.

Item #1 was the coronal suture part; item #6, the sagittal suture part; item #8, the squamous suture part; and item #9, the lambdoid suture part; were areas at which three points were treated.

Item #3, the spenofrontal suture part; item #5, the frontozygomatic suture part; item #7, the spenoparietal suture part; item #10, the occipitomastoid suture part; item #11, the spenosquamous suture part; and item #13, the temporozygomatic suture part; were areas at which one point was treated.

Items #2, #4, #12, and #14 are not applicable for cranial suture and were treated at one point.

With regard to the points of evaluation for stages 1, 2, and 3 of this cranial suture traction therapy program, whose validity and homogeneity (credibility) were verified, item #4 was deleted from the 14 items after it failed to meet the criteria of evaluation. Thus, 13 items were evaluated.

The categorization of each item in this study can be found in some manual skin care treatment therapies that use similar methods. However, there has never been an authorized care program that involves the cranial suture traction therapy based on scientific analyses and data, as well as one that involves the correction of facial asymmetry through cranial suture traction therapy [57,58,59]. The program will be used in the dental and medical world, as well as in some skin care academies and associations, to theorize the correction of facial asymmetry.

This study shows that it is possible to change the shape of the face by applying pressure to the cranium for correcting facial asymmetry. Our study also found that if plagiocephaly (of the cranial’s soft tissues) is treated by applying pressure to parts of the cranium, it is possible to prevent facial misalignment or asymmetry that would otherwise have ensued in cases where facial asymmetry is caused by one’s environment after birth. This would allow one to retain an agreeable face shape.

Antonio Di Ieva et al. [60] mentioned that suture lines follow a clearer pattern until 40 years of age, following which significant changes occur, such as rapid closing or invisible suture lines, depending on the individual. While interpreting these results, cranial suture traction therapy was considered effective between 20 years to 40 years of age.

Accordingly, Park [61] scientifically proved “The effect of cranial suture traction therapy on hard and soft tissue alignment with facial asymmetry in women in their 20s” through experimental studies. Subsequently, the therapy was effective at the hard tissue Sojv-Cg-Ans, Mx-Cg-ANS, and Go-ANS-Me angles, and soft tissue Ala-M-Sn and Ch-Sn-Me’ angles before and after 4 weeks of the procedure owing to the post-verification of hard and soft tissues. Following 4 weeks and 8 weeks of the procedure and 2 weeks after continuity confirmation, the effect was only observed at the Ex-M-Sn and Ala-M-Sn angles of soft tissues.

There are some limitations to this study. First, the results cannot be generalized as they were obtained by 15 experts. Second, a verification process is required to organize program development and apply the programs to various topics. This necessitates performing accurate measurements using valid inspection tools, such as cone beam [62] and panoramic radiography [63], which should be addressed in future studies.

## 5. Conclusions

The cranial suture traction therapy program will be useful for improving facial asymmetry and can be used as a supplementary therapy or special care therapy to help achieve balance. It has validity and credibility as a program that can help correct facial asymmetry by applying gentle but prolonged force on the suture parts of the cranium through traction therapy. Its commercialization will contribute to the correction and prevention of facial misalignment or asymmetry. Finally, this therapy developed will be referred to as cranial suture traction therapy (CSTT).

## Figures and Tables

**Table 1 medicina-58-00869-t001:** Expert panel composition.

Group	Stage 1	Stage 2	Stage 3
Rehabilitation medicine specialists, oriental medical doctors, dentistry specialists	6	6	6
Master’s or doctoral degree holders in skin care and cosmetology with more than 10 years of experience in the field	5	5	5
Educators in the field of skin care	4	4	4
Total	15	15	15

**Table 2 medicina-58-00869-t002:** Fourteen selected items according to the development test of the cranial suture traction therapy program.

Traction Therapy Item	Question	Traction Therapy Item	Question
Frontal and parietal bone traction	1	Parietal and temporal bone traction	1
Frontal and maxillary bone traction	1	Parietal and occipital bone traction	1
Frontal and sphenoid bone traction	1	Occipital and temporal bone traction	1
Frontal and nasal bone traction	1	Sphenoid and temporal bone traction	1
Frontal and zygomatic bone traction	1	Sphenoid and zygomatic bone traction	1
Parietal and parietal bone traction	1	Zygomatic and temporal bone traction	1
Parietal and sphenoid bone traction	1	Zygomatic and maxillary bone traction	1
Total		14	

**Table 3 medicina-58-00869-t003:** Cranial suture traction therapy program (14 items).

Traction Therapy Item	Treatment Photograph	Item Description
1. Frontal and parietal bone traction (both sides)	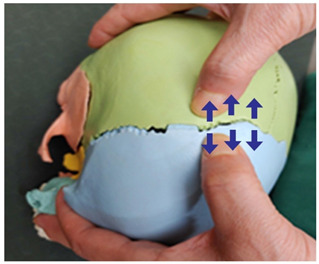	Therapist: supine position Hand position: From the center of the edge of the head, go up three fingerbreadths, and move and fix the thumbs at a 1 cm distance towards the front and the back of the line, one on the parietal bone side of the coronal suture and the other on the frontal side. Method: For the thumb on the frontal bone, exerting strength against the resisting periosteum, perform traction away from the thumb fixed on the parietal bone in the directions indicated by the arrows (3 points).
2. Frontal and maxillary bone traction (both sides)	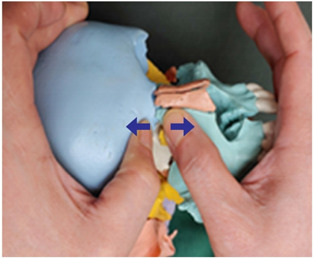	Therapist: supine position Hand position: Fix one hand on the superciliary arch that is on the inside of the frontal bone, and fix the other hand on the frontal process of the maxilla. Method: With the middle finger fixed on the superciliary arch that is on the inside of the frontal bone, and with the maxilla fixed in place by the other thumb, exerting strength against the resisting periosteum, perform traction in opposite directions indicated by the arrows (1 point).
3. Frontal and sphenoid bone traction (both sides)	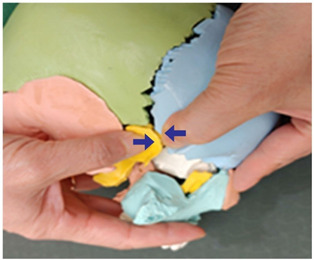	Therapist: lateral position Hand position: Drawing a line from the end of the eyebrow to the edge of the head, fix the thumbs on the outside and the inside of the middle point of that line. Method: At the middle point of the line, exert strength against the resisting periosteum using both thumbs and perform traction in the directions indicated by the arrows (1 point).
4. Frontal and nasal bone traction	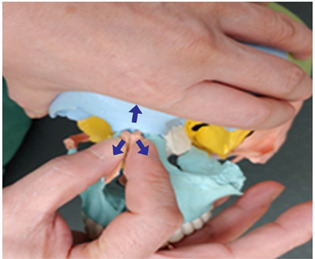	Therapist: supine position Hand position: Fix the thumb and the fingers on the frontal bones, while fixing the thumb and the index finger of the other hand on the nasal bone. Method: Perform traction towards the back with the palm that is fixed on both sides of the frontal bone and, when exhaling, perform downward traction of the nasal bone, exerting strength against the resisting periosteum (1 point).
5. Frontal and zygomatic bone traction (both sides)	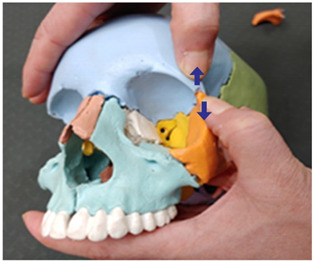	Therapist: supine position Hand position: Fix both thumbs at the recess that is at the zygomatic process of the frontal bone above the lateral canthus. Method: Perform traction with the thumbs in the directions indicated by the arrows, exerting strength against the resisting periosteum (1 point).
6. Parietal and parietal bone traction	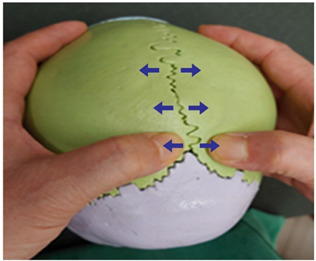	Therapist: prone position Hand position: Fix each of the thumbs at points marked by one finger breadth distance front and back from the center of the line that connects the two ears. Method: Perform traction with both thumbs from the middle point of the head towards the directions indicated by the arrows, exerting strength against the resisting periosteum (3 points).
7. Parietal and sphenoid bone traction (both sides)	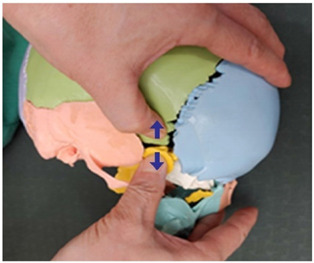	Therapist: lateral position Hand position: Drawing a line from the lateral canthus to the edge of the head that is parallel to the line connecting the end of the eyebrow to the edge of the head, fix both thumbs at the middle point of that line. Method: Fixing both thumbs at the middle point of the line connecting the lateral canthus to the edge of the head, perform traction in the directions indicated by the arrows, exerting strength against the resisting periosteum (1 point).
8. Parietal and temporal bone traction (both sides)	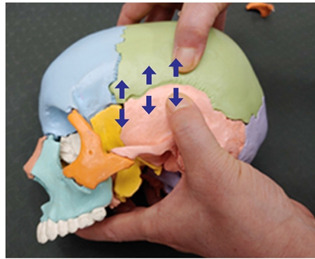	Therapist: lateral position Hand position: Find the point where the horizontal line drawn from the end of the eyebrow meets the vertical line drawn from the top of the ear, and fix each of the thumbs at points marked by one finger breadth distance front and back from that point. Method: Perform traction with the thumbs in the directions indicated by the arrows, exerting strength against the resisting periosteum (3 points).
9. Parietal and occipital bone traction (both sides)	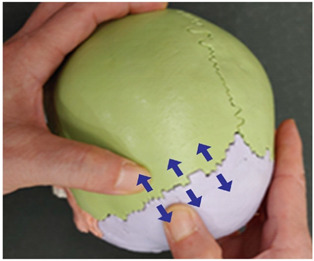	Therapist: prone position Hand position: At the 1/3 point of the horizontal line drawn from one ear to the other, fix each of the thumbs at points marked by one finger breadth distance front and back from that point. Method: Perform traction with the thumbs that are fixed at the 1/3 point in the directions indicated by the arrows (3 points).
10. Occipital and temporal bone traction (both sides)	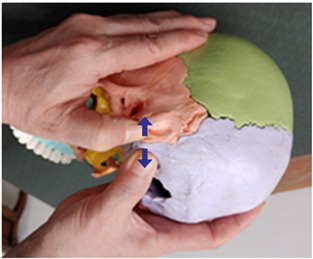	Therapist: prone position Hand position: Fix both thumbs at the recess that is at the back of the mastoid. Method: Perform traction with the thumbs in the directions indicated by the arrows, exerting strength against the resisting periosteum (1 point).
11. Sphenoid and temporal bone traction (both sides)	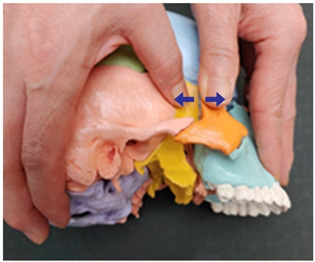	Therapist: lateral position Hand position: Fix both thumbs at the middle point of the sideburns line above the temporal process. Method: Perform traction with the thumbs in the directions indicated by the arrows, exerting strength against the resisting periosteum (1 point).
12. Sphenoid and zygomatic bone traction (both sides)	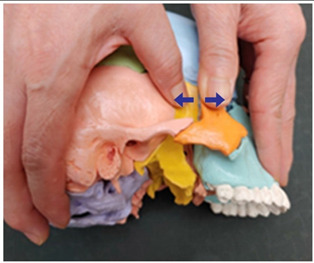	Therapist: lateral position Hand position: Following the horizontal line drawn from the lateral canthus, fix one thumb at the zygomatic bone and the other at the recess behind it. Method: Perform traction with the thumbs in the directions indicated by the arrows, exerting strength against the resisting periosteum (1 point).
13. Zygomatic and temporal bone traction (both sides)	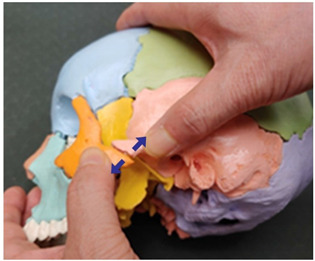	Therapist: lateral position Hand position: Fix both thumbs at the middle point of the zygomatic process from the temporal bone line above the temporal process. Method: Perform traction with the thumbs in the directions indicated by the arrows, exerting strength against the resisting periosteum (1 point).
14. Zygomatic and maxillary bone traction (both sides)	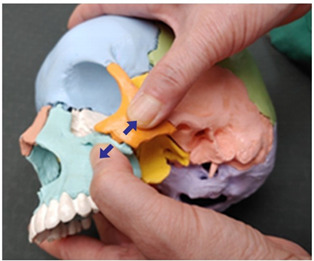	Therapist: lateral position Hand position: Fix one thumb at the mid point of the zygomatic process, and the other at the mid point of the maxilla. Method: Perform traction with the thumbs in the directions indicated by the arrows, exerting strength against the resisting periosteum (1 point).

**Table 4 medicina-58-00869-t004:** Treatment points according to the 14 items of cranial suture traction therapy.

Separation	Traction Therapy Item	Item Point
Coronal suture	Frontal and parietal bone traction	Both-sided procedure with three points
Not applicable for cranial suture	Frontal and maxillary bone traction	Both-sided procedure with one point
Sphenofrontal suture	Frontal and sphenoid bone traction	Both-sided procedure with one point
Not applicable for cranial suture	Frontal and nasal bone traction ※ #4 was deleted as a result of the opinions gathered from the experts during the 1st survey	One-point procedure
Frontozygomatic suture	Frontal and zygomatic bone traction	Both-sided procedure with one point
Sagittal suture	Parietal and parietal bone traction	Three-point procedure
Sphenoparietal suture	Parietal and sphenoid bone traction	Both-sided procedure with one point
Squamous suture	Parietal and temporal bone traction	Both-sided procedure with three points
Lambdoid suture	Parietal and occipital bone traction	Both-sided procedure with three points
Occipitomastoid suture	Occipital and temporal bone traction	Both-sided procedure with one point
Spenosquamous suture	Sphenoid and temporal bone traction	Both -sided procedure with one point
Not applicable for cranial suture	Sphenoid and zygomatic bone traction	Both -sided procedure with one point
Temporozygomatic suture	Zygomatic and temporal bone traction	Both-sided procedure with one point
Not applicable for cranial suture	Zygomatic and maxillary bone traction	Both-sided procedure with one point

※ #4: Refer to “Frontal and Nasal bone transaction” among the 14 items of “Table 3. Cranial strategy therapy program”.

**Table 5 medicina-58-00869-t005:** Analysis of the validity of the assessment factors of the results of the cranial suture traction therapy program in the second survey.

Questions Remaining in the First Survey	Descriptives	Central Tendency			CVR	Q1	Q3	Median	Consensus	Convergence
	Second Survey (M ± SD)	Median	Mode	IQR						
1	4.40 ± 0.51	4	4	4–5	1.0	4	5	4	0.75	0.50
2	4.67 ± 0.49	5	5	4–5	1.0	4	5	5	0.80	0.50
3	4.60 ± 0.58	5	5	4–5	1.0	4	5	5	0.80	0.50
5	4.53 ± 0.52	5	5	4–5	1.0	4	5	5	0.80	0.50
6	4.53 ± 0.52	5	5	4–5	1.0	4	5	5	0.80	0.50
7	4.47 ± 0.52	4	4	4–5	1.0	4	5	4	0.75	0.50
8	4.53 ± 0.52	5	5	4–5	1.0	4	5	5	0.80	0.50
9	4.53 ± 0.52	5	5	4–5	1.0	4	5	5	0.80	0.50
10	4.53 ± 0.52	4	5	4–5	1.0	4	5	5	0.80	0.50
11	4.53 ± 0.52	5	5	4–5	1.0	4	5	5	0.80	0.50
12	4.67 ± 0.49	5	5	4–5	1.0	4	5	5	0.80	0.50
13	4.67 ± 0.49	5	5	4–5	1.0	4	5	5	0.80	0.50
14	4.67 ± 0.49	5	5	4–5	1.0	4	5	5	0.80	0.50

1. Frontal and parietal bones; 2. Frontal and maxillary bones; 3. Frontal and sphenoid bones; 5. Frontal and zygomatic bones; 6. Parietal and parietal bones; 7. Parietal and sphenoid bones; 8. Parietal and temporal bones; 9. Parietal and occipital bones; 10. Occipital and temporal bones; 11. Sphenoid and temporal bones; 12. Sphenoid and zygomatic bones; 13. Zygomatic and temporal bones; and 14. Zygomatic and maxillary bones. IQR, Inter Quartile Range; CVR, content validity ratio.

**Table 6 medicina-58-00869-t006:** Analysis of the validity of the assessment factors of the results of the cranial suture traction therapy program in the third survey.

Questions Remaining in the First Survey	Descriptives	Central Tendency			CVR	Q1	Q3	Median	Consensus	Convergence
	Third Survey (M ± SD)	Median	Mode	IQR						
1	4.87 ± 0.35	5	5	5–5	1.0	5	5	5	1.00	0.00
2	4.87 ± 0.35									
3	4.93 ± 0.26									
5	4.93 ± 0.26									
6	4.93 ± 0.26									
7	4.93 ± 0.26									
8	4.93 ± 0.26									
9	4.87 ± 0.35									
10	4.93 ± 0.26									
11	4.93 ± 0.26									
12	4.93 ± 0.26									
13	4.93 ± 0.26									
14	4.93 ± 0.26									

1. Frontal and parietal bones; 2. Frontal and maxillary bones; 3. Frontal and sphenoid bones; 5. Frontal and zygomatic bones; 6. Parietal and parietal bones; 7. Parietal and sphenoid bones; 8. Parietal and temporal bones; 9. Parietal and occipital bones; 10. Occipital and temporal bones; 11. Sphenoid and temporal bones; 12. Sphenoid and zygomatic bones; 13. Zygomatic and temporal bones; and 14. Zygomatic and maxillary bones. IQR, Inter Quartile Range; CVR, content validity ratio.

**Table 7 medicina-58-00869-t007:** Analysis of the homogeneity (credibility) of the assessment factors of the results of the cranial suture traction therapy program.

Questions Remaining in the First Survey	Second Survey (M ± SD)	Third Survey (M ± SD)	*p*
1. Frontal and parietal bones	4.40 ± 0.51	4.87 ± 0.35	0.756
2. Frontal and maxillary bones	4.67 ± 0.49	4.87 ± 0.35	0.591
3. Frontal and sphenoid bones	4.60 ± 0.58	4.93 ± 0.26	0.398
5. Frontal and zygomatic bones	4.53 ± 0.52	4.93 ± 0.26	0.333
6. Parietal and parietal bones	4.53 ± 0.52	4.93 ± 0.26	0.268
7. Parietal and sphenoid bones	4.47 ± 0.52	4.93 ± 0.26	0.333
8. Parietal and temporal bones	4.53 ± 0.52	4.93 ± 0.26	0.268
9. Parietal and occipital bones	4.53 ± 0.52	4.87 ± 0.35	0.155
10. Occipital and temporal bones	4.53 ± 0.52	4.93 ± 0.26	0.333
11. Sphenoid and temporal bones	4.53 ± 0.52	4.93 ± 0.26	0.333
12. Sphenoid and zygomatic bones	4.67 ± 0.49	4.93 ± 0.26	0.143
13. Zygomatic and temporal bones	4.67 ± 0.49	4.93 ± 0.26	0.464
14. Zygomatic and maxillary bones	4.67 ± 0.49	4.93 ± 0.26	0.464

## Data Availability

Data not available due to ethical restrictions. Due to the nature of this research, participants of this study did not agree to their data being shared publicly; therefore, the supporting data are not available.

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
