# Peer review of "Development of a Cranial Suture Traction Therapy Program for Facial Asymmetry Correction Using the New Delphi Technique"

_medicina, 2022, doi:10.3390/medicina58070869_

Round 1

Reviewer 1 Report

The manuscript is well-written and the topic extremely interesting. The methods are clearly described and the results properly presented. Importantly, the authors tried to validate an osteopathic technique, which at present is not strongly supported by literature evidence. As a minor comment, the authors are encouraged to report some quantifiable outcomes of the treatment in terms of reduction of the asymmetry, in order to enhance the validity of their results.

Author Response

Thank you for your encouraging comments. We have added the quantifiable outcomes of the treatment as requested.

Page 19, lines 525-530:

“Post-verification of hard tissue and soft tissue showed that therapy was effective at the hard tissue Sojv-Cg-Ans, Mx-Cg-ANS, Go-ANS-Me angles, and soft tissue Ala-M-Sn, Ch-Sn-Me' angles before and after 4 weeks of procedure. After 4 weeks of procedure, 8 weeks of procedure, and 2 weeks after continuity confirmation, the effect was only visible at the angles Ex-M-Sn and Ala-M-Sn of soft tissue.”

We have also added further information on the procedures followed to verify the validity and reliability of the Delphi research method used in this study.

Page 13, lines 27-289:

“The second survey reflected responses collected in the first survey and was a structured closed questionnaire developed through consultation with experts including a rehabilitation medicine specialist, a doctor from a Chinese medical college, a dental specialist, and two professors; of this group, four are experts with PhD degreees in their respective fields. Later, three advisors (the rehabilitation medicine specialist, the doctor from a Chinese medical college, and the professor from the Department of Skin Care and Cosmetology) constituting an expert group, further evaluated the validity of the questions using a 5-point Likert scale. In addition, if necessary, the structured contents of the first survey were modified as required.

The analysis results of the responses of the second Delphi survey were then discussed by the expert group to reach consensus on the questions to be presented during the third Delphi survey. The validity of each question was indicated by a 5-point Likert scale by the advisors as was done for the second Delphi survey.”

Reviewer 2 Report

The manuscript addresses the relevant topic of cranial suture traction therapy for asymmetry correction. The authors should address the following issues to improve the quality of the manuscript:

- Please specify the age range of applicability of the technique

- Please specify how asymmetry is determined. It could be useful to add diagnostic imaging as an additional tool. The use of Cone beam and panoramic radiography could be advised. Please check 10.3390/app11177858 and 10.1127/homo/2020/1063 for the value of imaging in assessing asymmetry and the structure of cranial sutures.

Author Response

Reply 1:

Thank you for your comments. We have added information and relevant references to the Discussion section on the age range for which this technique can be reliably used.

Page 19, lines 519-525:

“Di Ieva et al. [59] also suggested that suture lines follow a clearer pattern until age 40, but that afterward, significant changes occur, such as rapid closing or invisible suture lines, depending on the individual. Based on these results, it is thought that this technique will be effective for patients between the ages of 20 to 40. Similarly, Park [60] scientifically proved "the effect of cranial suture traction therapy on hard and soft tissue alignment with facial asymmetry in women in their 20s" through experimental studies.”

Reply 2:

Thank you for your comments. We have added the quantifiable outcomes of the treatment as requested.

Page 19, lines 525-530:

“Post-verification of hard tissue and soft tissue showed that therapy was effective at the hard tissue Sojv-Cg-Ans, Mx-Cg-ANS, Go-ANS-Me angles, and soft tissue Ala-M-Sn, Ch-Sn-Me' angles before and after 4 weeks of procedure. After 4 weeks of procedure, 8 weeks of procedure, and 2 weeks after continuity confirmation, the effect was only visible at the angles Ex-M-Sn and Ala-M-Sn of soft tissue.”

We have also added information on the use of Cone beam and panoramic radiography as imaging diagnostic tools to the Discussion section and have reviewed and used the references suggested.

Page 19, lines 532-535:

“Second, a verification process is needed for program development. Future studies should, therefore, further validate their results using diagnostic imaging tools such as cone beam [61] and panoramic radiography [62].”